# UniQ: Unified Decoder with Task-specific Queries for Efficient Scene Graph Generation

### Xinyao Liao
Huazhong University of Science and Technology
School of Computer Science and Technology, Cognitive
Computing and Intelligent Information Processing (CCIIP)
Laboratory
Wuhan, China
m202373829@hust.edu.cn

### Wei Wei*
Huazhong University of Science and Technology
School of Computer Science and Technology, Cognitive
Computing and Intelligent Information Processing (CCIIP)
Laboratory
Wuhan, China
weiw@hust.edu.cn

### Dangyang Chen
Pingan Technology
Shenzhen, China
dangyangchen0511@163.com

### Yuanyuan Fu*
Pingan Technology
Shenzhen, China
fuyuanyuan723@pingan.com.cn

## Abstract

Scene Graph Generation(SGG) is a scene understanding task aims at identifying object entities and reasoning their relationships within a given image. In contrast to prevailing two-stage methods based on a large object detector (e.g., Faster R-CNN), one-stage methods integrate a fixed-size set of learnable queries to jointly reason relational triplets <subject, predicate, object>. This paradigm demonstrates robust performance with significantly reduced parameters and computational overhead. However, the challenge in one-stage methods stems from the issue of weak entanglement, wherein entities involved in relationships require both coupled features shared within triplets and decoupled visual features. Previous methods either adopt a single decoder for coupled triplet feature modeling or multiple decoders for separate visual feature extraction but fail to consider both. In this paper, we introduce **UniQ**, a **Uni**fied decoder with task-specific **Q**ueries architecture, where task-specific queries generate decoupled visual features for subjects, objects, and predicates respectively, and unified decoder enables coupled feature modeling within relational triplets. Experimental results on the Visual Genome dataset demonstrate that **UniQ** has superior performance to both one-stage and two-stage methods.

## CCS Concepts

• **Computing methodologies** → **Scene understanding**.

## Keywords

Scene Graph Generation; Visual Relationship Detection; One-stage model

---
*Corresponding author.

**ACM Reference Format:**
Xinyao Liao, Wei Wei, Dangyang Chen, and Yuanyuan Fu. 2024. UniQ: Unified Decoder with Task-specific Queries for Efficient Scene Graph Generation. In *Proceedings of the 32nd ACM International Conference on Multimedia (MM '24), October 28–November 1, 2024, Melbourne, VIC, Australia.* ACM, New York, NY, USA, 10 pages. https://doi.org/3664647.3681542

## 1 Introduction

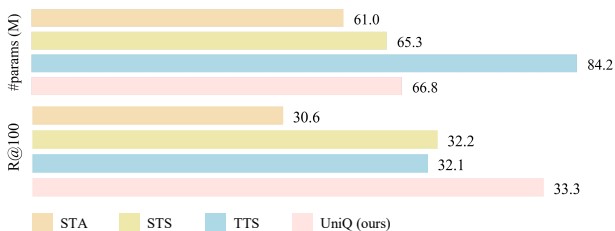

**Figure 1: Comparison of Baselines. We compare the number of parameters and Recall@100 among three baselines in Section 3 and our method UniQ. It demonstrates the formulation of STS baseline that we adopt in UniQ achieves better performance with fewer parameters.**

Scene Graph Generation (SGG) intends to form the semantic graph for a given image, where detected objects serve as nodes and relationships between them serve as edges. A relationship can equally be represented as a <subject, predicate, objected> triplet, and each element of the triplet is detailed by its class label and spatial location. Generating such scene graphs for a given image supports structured reasoning over high-level semantics, facilitating downstream complex tasks like visual question answering(VQA)[15, 47], image retrieval[19, 20], and image captioning[14, 58] with powerful representation.

Many works are proposed for the SGG problem, which can be roughly categorized into two classes, namely, **two**-stage based and **one**-stage based methods. The former usually employs a pre-trained object detector (eg. Faster R-CNN[39]) to generate *E* entity proposals. Then they pair each entity together as a complete graph for

relation inference, by exhausting all possible $E^2$ subject-object pairs. However, relying on a two-stage architecture may introduce the risk of error propagation, i.e., if some objects are omitted in the object detection stage, all relationships related to them can not be considered. One-stage methods pave the way for SGG by formalizing SGG as an end-to-end set prediction task, similar to DETR[1]. This architecture bypasses the subject-object paring step by decoding a fixed set of $N$ queries, successfully reducing the resolution space from $O(E^2)$ to $O(N)$. However, due to SGG having three inter-dependent sub-tasks: subject, object, and predicate prediction, the challenge of weak entanglement still persists in SGG. On the one hand, determining which entity plays a role as the subject or the object depends on features shared among relationships. On the other hand, locating and predicting entities rely on the decoupled visual features specific to each entity or relationship. Some previous one-stage methods[7, 23] harness multiple decoders to emit decoupled visual features separately for each sub-task but overlook modeling the coupled feature to enhance interactions within triplets, while other methods[56] learn holistic triplet features by single decoder, which are not able to model precise positional locations and decoupled semantic features specific to subjects, objects, and predicates.

In this work, we propose a new formulation through a **Uni**fied decoder with task-specific **Q**ueries, namely **UniQ**. UniQ treats each sub-task as decoding distinct sets of queries from a unified decoder rather than multiple decoders. The unified decoder takes three kinds of queries specific to subjects, objects, and predicates as input, and extracts decoupled visual features respectively for each sub-task in parallel. Different sets of task-specific queries undergo separate self-attention and cross-attention mechanisms with shared parameters. By sharing decoder parameters across sub-tasks, the decoder is enabled to concurrently generate features for subjects, objects, and predicates, facilitating mutual assistance among the sub-tasks. Besides utilizing task-specific queries to capture decoupled features for each sub-task, UniQ further enhances the interaction within triplets. In each decoder layer, UniQ fuses global and local context within triplets and embeds them into task-specific queries. This enables queries to dynamically capture detailed visual features according to the relational context. To acquire comprehensive global features, UniQ integrates subject, object, and predicate queries into a holistic query, thereby making every task-specific query perceive the entire triplet context. UniQ also employs a triplet self-attention module to capture the intricate interactions among subjects, objects, and predicates, thus extracting the local context in detail. The major contributions are summarized as follows:

- We propose a novel one-stage SGG formulation, UniQ, using a unified decoder with task-specific queries for parallel triplet decoding. This method enables detailed representation modeling for each sub-task and reduces parameter compared to methods that utilize multiple decoders (Figure 1).
- UniQ deals with the weak entanglement between entities and predicates by utilizing task-specific queries to model decoupled spatial locations and facilitating interaction within each triplet to model coupled semantic features.
- Extensive experimental results on the Visual Genome dataset demonstrate UniQ has superior performance to both one-stage and two-stage methods with fewer parameters.

## 2 Related Works

### 2.1 Scene Graph Generation

*2.1.1 Tow-stage Scene Graph Generation.* Researches on SGG continuously improve structured scene understanding. The prevailing two-stage methods decompose SGG into object detection and predicate classification tasks, where objects predicted from the first stage are exhaustively paired as input to the second stage. To construct visual context, several works employ Recurrent Neural Networks (LSTMs[13, 33, 52, 61], Tree-LSTMs[32, 44, 46], GRU[6, 51, 55]), Graph Neural Network[5, 11, 17, 43, 53, 54, 57], or attention mechanisms[21, 49, 66] to propagate information among entities and their relationships. Others leverage different kinds of prior knowledge, which are language prior[22], statistical prior[8, 61], and knowledge graph[3, 59]. Motifs [61] discovers that there are numerous motifs in SGG, indicating that predicate categories are largely determined by the labels of the subject-object pairs. [3] utilize a commonsense graph to predict visual relationships according to human instinct. The long-tail distribution of predicates existing in SGG datasets is a non-trivial problem. Loss re-weighting[2, 60], data augumention[10, 18, 31, 62], and unbiased representation learning[29, 37, 45, 50, 65] were used to mitigate this problem. However, two-stage methods are built upon an object detector with numerous parameters that result in substantial computations overhead and impede end-to-end training. Additionally, the two-stage methods pair all objects to generate triplets proposals, leading to a quadratic growth $O(E^2)$ in the solution searching space, where $E$ represents the number of object proposals.

*2.1.2 One-stage Scene Graph Generation.* Inspired by DETR[1], one-stage methods[7, 9, 23, 27, 30, 41, 48] predict all relationships at once by using an encoder-decoder architecture. These one-stage methods bypass pairing between all object proposals and directly decode <subject, predicate, object> triplets, which cost less parameters and computations. Integrating object detection and predicate prediction in the same step can present a quandary. While predicting the semantic label and spatial location of an entity may not necessitate any additional details from the triplet, identifying whether an entity corresponds to a specific relationship requires such information. PSGTR[56] utilizes a single decoder with holistic triplet queries, carrying features from different distributions (subject/object/predicate distributions), to predict the whole scene graph. These over-entangled features make it hard to model task-specific representations. PSGFormer[56], IterSGG[23], TraCQ[9] devise multiple decoders to extract less entangled visual features by decoding subjects, objects, and predicates separately. In contrast, multiple decoders make coupled features shared in triplets hard to extract. We construct a simple but effective architecture to tackle this issue. We leverage a unified decoder taking task-specific queries as input, which not only extracts decoupled visual features separately but also facilitates intricate interactions within triplets.

### 2.2 DETR and Its Variants

DETR[1] drops hand-designed components like spacial anchors[34] and non-maximal suppression[38] by adopting transformer encoder-decoder architecture[42] and a one-to-one assignment strategy[26]. But DETR suffers from low convergence issues and requires 500

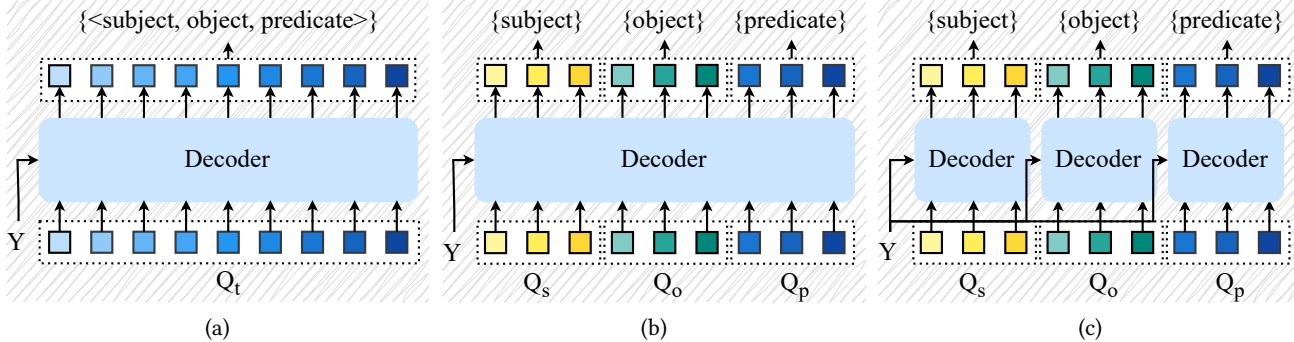

**Figure 2: Formulation of baselines. (a) Single decoder with task-agnostic queries: The single decoder takes triplet queries as input. Each query corresponds to predicting the whole triplet. (b) Single decoder with task-specific queries: The task-specific queries are input into a shared decoder. Each type of query responds to each sub-task. (c) Three decoders with task-specific queries: Three decoders separately predict each component of triplets.**

training epochs while Faster R-CNN[39] only needs 12 epochs in the COCO dataset[35]. Researchers[28, 63] find out that the instability of bipartite matching leads to slow convergence. They accelerate DETR training by using a denoising training module. Other works[4, 16, 67] demonstrate that the one-to-one assignment strategy generates too few positive samples, which hinders the model from learning discriminative features. In this paper, we follow Group DETR[4] to enable one-to-many assignments while training and speed up training efficiency.

## 3  Formulation

The task of SGG takes an image $\mathbf{I}$ as input. And then generates a scene graph $\mathbf{G}$ that consists of a set of relational triplets $< \mathbf{S}, \mathbf{P}, \mathbf{O} >$, where $\mathbf{S} = \{s_i\}_{i \leqslant n}$ denotes the set of subjects, $\mathbf{O} = \{o_i\}_{i \leqslant n}$ denotes the set of objects, $\mathbf{P} = \{p_i\}_{i \leqslant n}$ denotes the set of predicates, n denotes the number of triplets, $(s_i, p_i, o_i)$ represents the $i$-th triplet in $\mathbf{G}$. Therefore, the SGG task is equivalent to modeling the conditional distribution $P_r(\mathbf{G} \mid \mathbf{I})$. The two-stage methods decompose it into a product of conditionals as Equation (1). The first step aims to model $P_r(\mathbf{S}, \mathbf{O} \mid \mathbf{I})$ distribution, which means detects all entities of the given image and exhaustively pair them into subject-object pairs. $P_r(\mathbf{P} \mid \mathbf{S}, \mathbf{O}, \mathbf{I})$ means that predicate prediction is conditioned on the subject-object pairs detected and paired from the first stage. This formulation significantly relies on the ability of object detector due to the entities omitted in the first stage have no opportunity to serve as subjects or objects.

$$P_r(\mathbf{G} \mid \mathbf{I}) = P_r(\mathbf{S}, \mathbf{O} \mid \mathbf{I}) \cdot P_r(\mathbf{P} \mid \mathbf{S}, \mathbf{O}, \mathbf{I}) \qquad (1)$$

In contrast, one-stage methods jointly optimize both object detection and predicate prediction by an end-to-end architecture to circumvent the cascading errors that is commonly associated with two-stage processes. However, while two-stage methods explicitly predict predicate classes conditioned on subject-object pairs, one-stage methods call for finding a balance between decoupled features demanded by object detection and coupled features needed by predicate prediction.

We summarize two kinds of decoder structures in previous methods and propose a new formulation for better prediction.

**Single decoder with task-agnostic queries (STA) baseline**[56] uses a fixed set of task-agnostic queries as input for a single decoder as shown in Figure 2 (a). Compared to the formulation in Equation (1), it predicts not only subject-object pairs but also predicates accordingly at the same step as Equation (2). However, the subject, object, and predicate of a triplet occupy separate regions. Relying solely on a learned positional query may not be sufficient to precisely identify every component of triplets.

$$P_r(\mathbf{G} \mid \mathbf{I}) = P_r(\mathbf{S}, \mathbf{O}, \mathbf{P} \mid \mathbf{I}) \qquad (2)$$

**Three decoders with task-specific queries (TTS) baseline**[23, 24, 30] as shown in Figure 2 (c) devises three kinds of task-specific queries and input them into three decoders to predict subjects, objects, and predicates separately. As STA models subject, object, and predicate representations in the same query, coupled features shared within triplets are inherently contained in task-agnostic queries. For a fair comparison, we add an MLP before each decoder layer to model features shared among task-specific queries in the TTS and STS baselines.

**Single decoder with task-specific queries (STS) baseline** proposed by this paper is shown in Figure 2 (b). We unify three decoders into a decoder while taking three kinds of task-specific queries as input. This formulation not only enables generating triplets in parallel as Equation (2) but also models features specific to subjects, objects, and predicates. Our formulation of one-stage SGG can achieve competitive results with fewer parameters as shown in Figure 1.

## 4  Our Approach

Our proposed approach is composed of two major components: an image feature extractor implemented by a CNN backbone and a transformer encoder (Section 4.1), a relational triplet predictor implemented by a shared decoder with task-specific queries (Section 4.2). Like DETR[1], the image feature extractor first maps the original image to a condensed feature space with lower dimensions. The triplet predictor takes image features as input and directly predicts a set of triplets $< \mathbf{S}, \mathbf{O}, \mathbf{P} >$, eliminating the graph assembling module used in SGTR[30]. The approach follows the set prediction

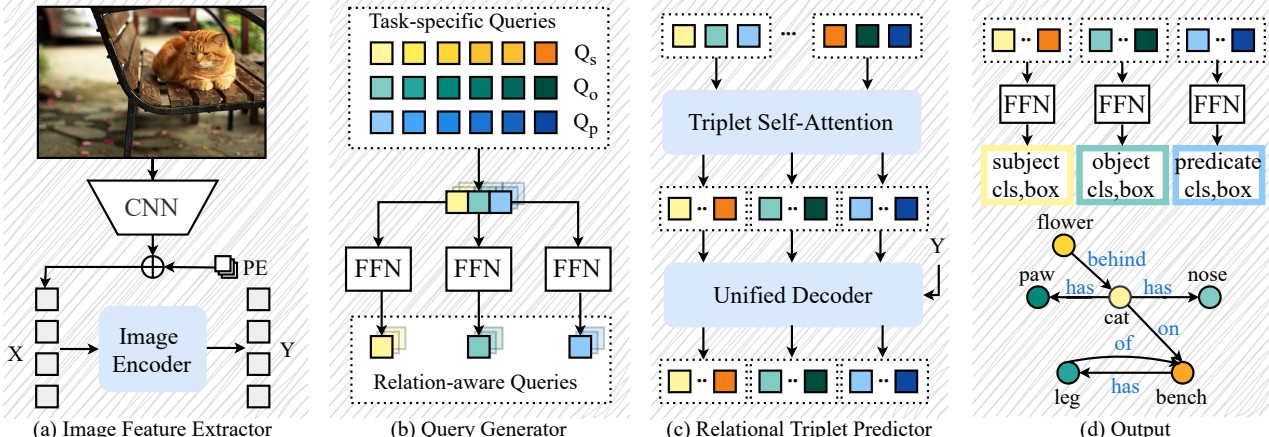

**Figure 3: Architecture Illustration. (a) Image Feature Extractor takes images as input and maps them to condensed image representations by a CNN backbone and a transformer encoder. (b) Query Generator depicts how to form task-specific relation-aware queries for decoding. (c) Relational Triplet Predictor has a triplet self-attention for capturing interaction within the triplet and a unified decoder for separately extracting visual features of each sub-task. (d) Output is generated by FFN.**

tradition as DETR[1] for end-to-end training and utilizes a one-to-many assignment strategy for faster convergence during training (Section 4.3). The framework of UniQ is represented in Figure 3.

## 4.1 Image Feature Extractor

For a given image $I \in \mathbb{R}^{3 \times H_0 \times W_0}$ that has 3 color channels, $H_0$ pixels in height and $W_0$ pixels in width, a conventional CNN backbone (eg. ResNet[12]) maps it into high-level image features $X \in \mathbb{R}^{C \times H \times W}$, where $C$ denotes the number of feature map channels, and $H$, $W$ correspond to the spatial dimensions of the lower-resolution map. A transformer encoder then flattens the spatial dimensions and extracts more compact features $Y \in \mathbb{R}^{d \times HW}$ with positional encodings $PE \in \mathbb{R}^{d \times HW}$ added to each layer. A $1 \times 1$ convolution reduces the dimension from $C$ to $d$. Equation (3) represents how the image feature extractor functions.

$$Backbone(I) \rightarrow X, \ Encoder(X) \rightarrow Y \tag{3}$$

## 4.2 Relational Triplet Predictor

Our Relational Triplet Predictor employs decoder architecture similar to Transformer[42] to decode the predicted relational triplets. The decoder takes three fixed-size sets of task-specific queries, i.e., subject queries $Q_s \in \mathbb{R}^{N \times d}$, object queries $Q_o \in \mathbb{R}^{N \times d}$, and predicate queries $Q_p \in \mathbb{R}^{N \times d}$ as input, and harnesses a parameter-shared decoder to generate task-specific representations all at once, denoted by $Z_x; x \in \{s, o, p\}$. The final step is to generate a set of triplet estimations $< \hat{S}, \hat{O}, \hat{P} >$ by input task-specific representations into feed-forward networks (FFN). The whole procedure is depicted as Equation (4).

$$Decoder(Q_s, \ Q_o, \ Q_p; Y) \rightarrow Z_s, \ Z_o, \ Z_p$$
$$FNN(Z_s) \rightarrow \hat{S}, \ FNN(Z_o) \rightarrow \hat{O}, \ FNN(Z_p) \rightarrow \hat{P} \tag{4}$$

The relational triplet predictor contains three components: (1) **Relation-aware Task-specific Queries**: This component firstly generates three sets of task-specific queries for subjects, objects,

and predicates respectively, then fuse each relational triplet to let the task-specific queries be aware of which relations they belong to (Section 4.2.1). (2) **Triplet Coupled Self-Attention**: This component operates a self-attention mechanism to model mutual interactions within triplet, i.e. how subject/object/predicate's prediction affect each other (Section 4.2.2). (3) **Decoupled Parallel Decoding**: This component captures the contexts of subjects, objects, and predicates respectively via a self-attention operation and extracting visual features from image representations in parallel via a cross-attention operation to model decoupled features specific to each sub-task (Section 4.2.3).

*4.2.1 Relation-aware Task-specific Queries.* We devise three types of task-specific queries for detailed triplet representation, They are three sets of learned embeddings $Q_s, Q_o, Q_p$, each set has $N$ queries with size $d$ of representations. Since $< q_{s,i}, q_{o,i}, q_{p,i} >$ represents the $i$-th triplet, each task-specific query necessitates to know which relationship it belongs to before separate decoding. We adopt a multi-layer perceptron (MLP) concatenating queries output from the previous $(l-1)$-th decoder layer to make them aware of their global relational contexts. This module can be formulated as:

$$Q_s^l = Q_s^{l-1} + MLP([Q_s^{l-1}; \ Q_o^{l-1}; \ Q_p^{l-1}]) \tag{5}$$

where $[ ; ]$ denotes the concatenate operation, $l$ denotes the $l$-th decoder layer. object queries $Q_o$ and predicate queries $Q_p$ can also be updated as Equation (5).

*4.2.2 Coupled Triplet Self-Attention.* This module aims to capture the detailed dependencies among triplet components. For instance, when provided with a relational triplet such as <human, ride, horse>, the spatial location of 'human' may assist in locating the object 'horse'. At a particular decoder layer $l$, $Q_s^l, Q_o^l, Q_p^l$ has the size of $(bs, N, d)$, where $bs$ is the number of batch size. To organize each triplet as a sequence, we reshape task-specific queries to the dimensions of $(1, bs \times N, d)$, subsequently concatenating them into triplet queries $Q_t^l$ with dimensions $(3, bs \times N, d)$. After that, the

self-attention mechanism successfully operates within each triplet, and explicitly models interactions of spatial and semantic information between different sub-tasks. As the transformer architecture maintains permutation invariance, positional encodings $PE_s^l$, $PE_o^l$, and $PE_p^l$, which share the same shape as queries, undergo a similar reshaping process as queries to form $PE_t^l$ and then added into the input of each attention layer. This module can be formulated as,

$$\mathbf{Q}_t^l = Attention(q = \mathbf{Q}_t^l + \mathbf{PE}_t^l, \ k = \mathbf{Q}_t^l + \mathbf{PE}_t^l, \ v = \mathbf{V}) \quad (6)$$

where $Attention(.)$ denotes a multi-head attention operation, $d$ denotes the feature dimension of $\mathbf{Q}$, $\mathbf{K}$, and $\mathbf{V}$, $\mathbf{Q}_t^l$ on the left side of equation denotes the triplet queries embedded with mutual information among subjects, objects, and predicates. $\mathbf{Q}_t^l$ then transits back to $\mathbf{Q}_s^l \in \mathbb{R}^{bs \times N \times d}$, $\mathbf{Q}_o^l \in \mathbb{R}^{bs \times N \times d}$, and $\mathbf{Q}_p^l \in \mathbb{R}^{bs \times N \times d}$, following the construction method in reverse.

*4.2.3 Decoupled Parallel Decoding.* After embedding coupled triplet features for each task-specific query, this module adheres to the conventional architecture of the transformer decoder. It performs self-attention and cross-attention operations over task-specific queries in parallel and predicts subjects, objects, and predicates respectively. To facilitate parallel decoding, task-specific queries $\mathbf{Q}_s^l$, $\mathbf{Q}_o^l$, and $\mathbf{Q}_p^l$ are concatenated along the batch size dimension. The above implementation enables task-specific queries to separately reason about the visual representations they require while using shared parameters. This module can be represented as,

$$\mathbf{Q}_s^l = Attention(q = \mathbf{Q}_s^l + \mathbf{PE}_s^l, \ k = \mathbf{Q}_s^l + \mathbf{PE}_s^l, \ v = \mathbf{Q}_s^l)$$
$$\mathbf{Q}_s^l = Attention(q = \mathbf{Q}_s^l + \mathbf{PE}_s^l, \ k = \mathbf{Y} + \mathbf{PE}_y, \ v = \mathbf{Y}) \quad (7)$$

where $\mathbf{Y}$ denotes image features output from the encoder as Equation (3), $\mathbf{PE}_y$ denotes the position encodings of the image feature map $\mathbf{Y}$. $\mathbf{Q}_o^l$ and $\mathbf{Q}_p^l$ are concurrently updated with shared parameters, similar to $\mathbf{Q}_s^l$.

## 4.3 End-to-end Training and One-to-many Assignment

Inspired by DETR[1], we redefine SGG as a set prediction task to facilitate end-to-end training, where we aim to generate a fixed-size set of $N$ triplet predictions represented as $\{(\hat{s}_i, \hat{o}_i, \hat{p}_i)\}_{i \leqslant N}$, with $\hat{s}_i = \{\hat{c}_s, \hat{b}_s\}$, $\hat{o}_i = \{\hat{c}_o, \hat{b}_o\}$, and $\hat{p}_i = \{\hat{c}_p, \hat{b}_p\}$. $\hat{c}_x; \mathbf{x} \in \{\mathbf{s}, \mathbf{o}, \mathbf{p}\}$ denotes the predicted class label and $\hat{b}_x; x \in \{s, o, p\}$ denotes the predicted bounding box. These predictions are to be matched with the ground-truth scene graph $\mathbf{G} = \{(s_i, o_i, p_i)\}_{i \leqslant m}$, where $m$ represents the number of ground-truth triplets. $N$ is always larger than $m$.

The triplet matching cost between a triplet prediction $(\hat{s}_i, \hat{o}_i, \hat{p}_i)$ and a ground-truth triplet $(s_i, o_i, p_i)$ consists of the cost of its subject, object, and predicate predictions as Equation (8).

$$C_{t,i} = C(\hat{s}_i, s_i) + C(\hat{o}_i, o_i) + C(\hat{p}_i, p_i)$$
$$C(\hat{x}_i, x_i) = \hat{p}(c_x) + \mathcal{L}_1(\hat{b}_x, b_x) + \mathcal{L}_{GIOU}(\hat{b}_x, b_x) \quad (8)$$

Where $i$ denotes the $i$-th triplet, $x \in \{s, o, p\}$. And the cost function $C$ is determined by the predicted class probability $\hat{p}$ and $L_1$ loss $\mathcal{L}_1$, generalized IOU loss[40] $\mathcal{L}_{GIOU}$ of bounding box.

Considering the number of ground truths $m$ is less than $N$, we pad the scene graph $\mathbf{G}$ with $\varnothing$ (no relation). The Hungarian algorithm is used to find a bipartite matching between two sets that has the lowest cost as Equation (9). $\sigma \in \mathfrak{S}_N$ is a permutation of $N$ elements. Instead of performing bipartite matching after $l$-th decoder layer akin to the original DETR[1], we aggregate costs across all auxiliary decoder layers for a single, stable matching pass as [23]. This approach consolidates the set of predictions from each decoder layer into a unified decision-making process.

$$\hat{\sigma} = \arg\min_{\sigma \in \mathfrak{S}_N} \sum_l \sum_i^N C_t \quad (9)$$

The triplet losses $\mathcal{L}_t$ can be represented as $\mathcal{L}_t = \mathcal{L}_s + \mathcal{L}_o + \mathcal{L}_p$. Given the bipartite matching $\hat{\sigma}$, $\mathcal{L}_s$, $\mathcal{L}_o$, and $\mathcal{L}_p$ are computed as Equation (10), Where $x \in \{s, o, p\}$, $\mathcal{L}_{box}$ is a linear combination of $\mathcal{L}_1$ loss and generalized IOU loss $\mathcal{L}_{GIOU}$. Due to the long-tail distribution of predicates, we conduct both biased and unbiased training for our method. For unbiased training, we adopt a re-weighted loss as [23], where the class weight of each predicate $c$ is $w_c = \max\{(\frac{\alpha}{f_c})^\beta, 1.0\}$, $f_c$ denotes the frequency of $c$ in training set, $\alpha$ and $\beta$ are hyper-parameters.

$$\mathcal{L}_x = \sum_i^N [-\log \hat{p}_{\hat{\sigma}(i)}(c_{x,i}) + \mathbb{1}_{\{c_{x,i} \neq \varnothing\}} \mathcal{L}_{box}(\hat{b}_{x,\hat{\sigma}(i)}, b_{x,i})] \quad (10)$$

Given that bipartite matching based one-to-one assignments produce a limited number of positive samples, thereby restricting model learning of generalized and discriminative representations, we adopt the one-to-many paradigm similar to Group DETR[4] for triplet assignments. We employ $K$ groups of $N$ learned task-specific queries, denoted as $\{\mathbf{Q}_{x,1}, \mathbf{Q}_{x,2}, \ldots, \mathbf{Q}_{x,K}\}$ for $x \in \{s, o, p\}$, which correspondingly yield $K$ groups of triplet predictions. We conduct one-to-one assignments on each based on the cost matrix, resulting in $K$ times matching $\{\hat{\sigma}_1, \hat{\sigma}_2, \ldots, \hat{\sigma}_K\}$ for each ground-truth triplet.

## 5 Experiments

### 5.1 Settings

*5.1.1 Dataset.* Our methods are trained and evaluated on Visual Genome (VG)[25], a large dataset featuring 108,077 structured images. Considering the extreme long-tail predicate distribution of VG, we adopt a frequently employed subset VG150 that contains the most frequent 150 object classes and 50 predicate classes. Following standard practice in previous works[55], we allocate 70% of images for training and reserve the remaining 30% for testing.

*5.1.2 Evaluation Metrics.* To evaluate our methods, we employ widely used class-agnostic metrics: recall@K (R@K) to assess the performance of predominant classes and mean recall@K (mR@K) to present the performance of tail classes. We also introduce harmonic racall@K (hR@K), introduced by [23] to show the overall improvement of recall and mean recall. Additionally, we utilize zero-shot recall@K (zs-R@K) to exhibit the model's generalization ability to unseen categories, and no-graph constraint recall@K (ng-R@K) to evaluate all possible predicates related to subject-object pairs. Note that, K= {20, 50, 100}. We also present the average precision AP@50 (IoU=0.5) to evaluate the quality of entities involved in relationships.

**Table 1: Scene Graph Generation Compared to Existing Methods. B denotes the type of backbone, and D denotes the type of detector. † means evaluate methods with top-3 links followed by [23][30] for fair comparison. ◇ indicates utilizing reweighted loss for unbiased training, and the scaling parameter $\alpha$ and $\beta$ is set to be $0.07$ and $0.75$ respectively. Red denotes the best performance of one-stage models, blue denotes the best performance of two-stage models, and underline denotes UniQ (ours) achieves the best performance on both one-stage and two-stage models.**

| B | D | Method | AP$_{50}$ | Mean Recall (↑) | | | Recall (↑) | | | Harmonic Recall (↑) | | | #Params (↓) |
|---|---|--------|-----------|------|------|------|------|------|------|------|------|------|------|
| | | | | @20 | @50 | @100 | @20 | @50 | @100 | @20 | @50 | @100 | (M) |
| X101-FPN | Two-stage based | IMP[55] | - | 2.8 | 4.2 | 5.3 | 18.1 | 25.9 | 31.2 | 4.8 | 7.2 | 9.1 | 203.8 |
| | | MOTIFS[61] | 20.0 | 4.1 | 5.5 | 6.8 | 25.1 | 32.1 | 36.9 | 7.0 | 9.4 | 11.5 | 240.7 |
| | | RelDN[64] | - | - | 6.0 | 7.3 | - | 31.4 | 35.9 | - | 10.1 | 12.1 | 615.6 |
| | | VCTree[46] | 28.1 | 5.4 | 7.4 | 8.7 | 24.5 | 31.9 | 36.2 | 8.8 | 12.0 | 12.1 | 360.8 |
| | | GPS-Net[36] | - | - | 6.7 | 8.6 | - | 31.1 | 35.9 | - | 11.0 | 13.9 | - |
| | | G-RCNN[57] | 24.8 | - | 5.8 | 6.6 | - | 29.7 | 32.8 | - | 9.7 | 11.0 | - |
| | | MOTIFS+TDE[45, 61] | 20.0 | 5.8 | 8.2 | 9.8 | 12.4 | 16.9 | 20.3 | 7.9 | 11.0 | 13.2 | 240.7 |
| | | MOTIFS+GCL[10, 61] | - | - | 16.8 | 19.3 | - | 18.4 | 22.0 | - | 17.6 | 20.6 | 240.7 |
| | | VCTree+TDE[45, 46] | 28.1 | 6.9 | 9.3 | 11.1 | 14.0 | 19.4 | 23.2 | 9.2 | 12.6 | 15.0 | 360.8 |
| | | VCTree+GCL[10, 46] | - | - | 15.2 | 17.5 | - | 17.4 | 20.7 | - | 16.2 | 18.9 | 360.8 |
| | | CV-SGG[18] | - | - | 14.8 | 17.1 | - | 27.8 | 32.0 | - | 19.2 | 22.0 | - |
| | | BGNN[31] | 29.0 | 7.5 | 10.7 | 12.6 | 23.3 | 31.0 | 35.8 | 11.3 | 15.9 | 18.6 | 341.9 |
| ResNet-101 | One-stage based | HOTR[24] | - | - | 9.4 | 12.0 | - | 23.5 | 27.7 | - | 13.4 | 16.7 | - |
| | | Relationformer[41] | 26.3 | 4.6 | 9.3 | 10.7 | 22.2 | 28.4 | 31.3 | 8.0 | 14.0 | 16.0 | 92.9 |
| | | SGTR†[30] | - | - | 12.0 | 15.2 | - | 24.6 | 28.4 | - | 16.1 | 19.8 | 166.5 |
| | | IterSG[23] | - | - | 8.0 | 8.8 | - | 29.7 | 32.1 | - | 12.6 | 13.8 | 93.2 |
| | | IterSG†◇[23] | 27.7 | 11.3 | 16.7 | 21.4 | 19.7 | 28.5 | 34.3 | 14.4 | 21.1 | 26.4 | 93.2 |
| | | **UniQ (ours)** | 28.6 | 6.3 | 8.5 | 9.6 | 25.2 | 30.5 | 33.2 | 10.2 | 13.3 | 14.9 | 66.8 |
| | | **UniQ†◇ (ours)** | 28.4 | 11.3 | 16.8 | 21.1 | 20.1 | 30.0 | 36.2 | 14.5 | 21.5 | 26.7 | 66.8 |

## 5.2 Implementation Details

We utilize ResNet-101[12] as the backbone network and a 6-layer transformer in the image feature extractor component.(Section 4.1). A 6-layer decoder with 8 attention heads and feature size of 256 is used for relational triplet prediction. The number of queries $N$ is 300 and the number of query group $K$ is 3. The FFNs for relational-aware queries have 3 linear layers with ReLU. (Section 4.2)

We use pre-trained parameters of DETR[1] for object detection on VG as previous works[9, 23, 30] to speed up the convergence. For training, the initial learning rate of the backbone and the encoder-decoder architecture are set to be $10^{-5}$ and $10^{-4}$ respectively with the optimizer ADAMW. We train UniQ and baselines on 4 RTX 4090 GPUs with a batch size of 12.

## 5.3 Comparisons to Existing Methods

As shown in Table 1, we compared our method with both two-stage methods based on Faster R-CNN[39] ([10, 18, 31, 36, 45, 46, 55, 57, 61, 64]) and one-stage methods based on DETR[1] ([23, 24, 30, 41]). As for the one-stage methods comparison, our proposed UniQ outperforms the state-of-art method IterSG[23] with a margin of **0.5/0.8** and **0.8/1.1** on mean recall@50/100 and recall@50/100. SGTR[30] and IterSG[23] adopt a top-k strategy to select more possible candidate triplets. We reimplement the strategy by selecting the 3 most likely predicates for each entity pair to evaluate our model. The result shows that UniQ†◇ surpasses IterSG[23] and SGTR[30] on recall@100 (**1.9** and **7.8**). UniQ†◇ also achieves the best performance on harmonic recall@20/50/100. In conclusion, UniQ achieves

**Table 2: Results of zs-R@K and ng-R@K on Visual Genome.**

| Methods | zs-R | | ng-R | |
|---------|------|------|------|------|
| | @50 | @100 | @50 | @100 |
| MOTIFS[61] | 0.1 | 0.2 | 30.5 | 35.8 |
| BGNN[31] | 0.4 | 0.9 | - | 32.2 |
| Relationformer[41] | - | - | 31.2 | 36.8 |
| IterSG[23] | 2.7 | 3.8 | 30.5 | 35.5 |
| **UniQ (ours)** | **2.8** | **3.9** | **34.0** | **38.6** |
| **UniQ◇ (ours)** | **3.2** | **4.5** | **32.1** | **37.6** |

considerable performance improvement with at least **28%** fewer parameters in comparison to existing one-stage methods that utilize ResNet-101[12] as the backbone. To be compared with two-stage models that have a heavy object detector and always use extra features, our method still outperforms most of the two-stage methods. [36, 46, 55, 57, 61, 64] are tow-stage methods without unbiased training strategy, while [10, 18, 31, 45] are tow-stage methods using unbiased strategies. Our method UniQ†◇ employs the reweighted loss strategy that can compete with all the unbiased two-stage methods. Our method UniQ without biased training achieves the highest recall@20 among all methods.

We also report zero-shot recall (zs-R@50/100) and no-graph con-straint recall (ng-R@50/100) results as shown in Table 2. We select methods that demonstrate good performance on recall and mean

**Table 3: Ablation on task-specific queries. STA: Single decoder with task-agnostic queries. STS: Single decoder with task-specific queries. TTS: Three decoders with task-specific queries.**

| Baselines | $AP_{50}$ | R@20 | R@50 | R@100 | #params (M) |
|-----------|-----------|------|------|-------|-------------|
| STA | 27.0 | 22.5 | 27.6 | 30.6 | 61.0 |
| STS | 28.6 | 23.3 | 28.9 | 32.2 | 65.3 |
| TTS | 28.8 | 23.2 | 28.7 | 32.1 | 84.2 |

**Table 4: Ablation on model components. RQ denotes relation-aware queries, TSA denotes triplet self-attention.**

| # | RQ | TSA | R@20 | R@50 | R@100 | #params (M) |
|---|----|----|------|------|-------|-------------|
| 1 | ✓ | ✓ | 23.3 | 29.4 | 33.3 | 66.8 |
| 2 |   |   | 22.0 | 26.9 | 30.0 | 61.5 |
| 3 | ✓ |   | 23.3 | 28.9 | 32.2 | 65.3 |
| 4 |   | ✓ | 23.1 | 28.3 | 31.5 | 62.7 |

recall as baselines. MOTIFS[61] and BGNN[31] represent the performance of the biased and unbiased two-stage method respectively. Relationformer[41] and IterSG[41] represent the performance of the biased and unbiased one-stage method respectively. The experimental results exhibit that our approach achieves the best performance both on zs-R@K and ng-R@K, proving that UniQ has better generalization ability compared to existing methods. Our biased training method surpasses MOTIFS[61] and Relationformer[41] **2.8** and **1.8** on ng-R@100, **3.5** and **2.8** on ng-R@50 respectively. Our unbiased training method competes with IterSG[23] and sees performance improvement on all metrics.

## 5.4 Ablation Studies

*5.4.1 Ablation on task-specific queries to prove the effectiveness of decoupled features.* We conduct experiments on three baselines to prove that task-specific queries model subjects, objects, and predicates in detail and achieve better performance with the tiny amount of parameters increasing. STA denotes the baseline only has a decoder and one type of query to predict the whole triplet. STA denotes the method we adopt, which inputs task-specific queries into a shared decoder. TTS denotes using three decoders with distinct sets of queries to decode subjects, objects, and predicates separately. The experiment results are reported in Table 3, that STS and TTS baselines achieve better performance compared to STA baseline, which demonstrates the effectiveness of task-specific queries. Furthermore, STS baseline can compete with the TTS baseline with almost **25%** reducing of parameters, which conveys that multiple decoders are almost redundant.

*5.4.2 Ablation on Model Components to prove the effectiveness of coupled features.* Besides incorporating task-specific queries to model decoupled features of subjects, objects, and predicates. Our proposed model further probes the modeling of coupled features shared within triplets. Relation-aware queries (RQ) are implemented before each decoder layer to fuse features within the

**Table 5: Ablation on the the hyper-parameter $K$. $K$ denotes the number of query groups in the decoder.**

| K | mR@50/100 | R@50/100 | zs@50/100 |
|---|-----------|----------|-----------|
| 1 | 14.7/16.3 | 27.1/30.1 | 2.8/3.6 |
| 3 | 14.3/16.8 | 29.4/33.3 | 3.2/4.5 |
| 5 | 14.4/16.4 | 29.2/32.6 | 3.2/4.3 |

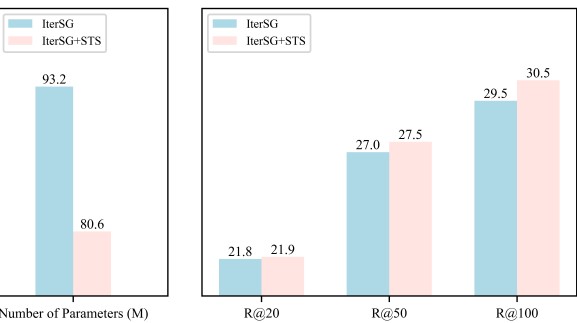

**Figure 4: Ablation on the transferability of the STS paradigm.**

triplet, enabling the perception of the global triplet context for each sub-task. As shown in Table 4, experiments 2 and 3 demonstrate the effectiveness of RQ with the average enhancement of **4.2** recall rate. Triplet Self-attention (TSA) is aimed at boosting interaction among triplets by modeling their influence on each other. experiments 3 and 4 demonstrate the effectiveness of TSA, showing that with only 1.3M parameters increasing, the Recall@20 sees a rise of **2.2**. Table 4 proves that capturing the semantic information within each triplet assists the model for relational reasoning.

*5.4.3 Ablation on hyper-Parameter $K$.* We employ a one-to-many assignment strategy by initializing multiple query groups to stabilize bipartite matching and augment positive samples. The ablation experiment on the number of query groups is conducted as Table 5. The performance increases when the number of query groups grows. While the number of query groups reaches 5, the performance becomes stable. So we adopt $K = 3$ in all experiments. Note that the mean Recall metrics do not see an obvious improvement when training with more groups. This result may be due to the data augmentation that is brought by multiple query groups is dominant by head categories in distribution.

*5.4.4 Ablation on the transferability of the STS paradigm.* We apply the STS paradigm on IterSG[23] model to prove its transferability, i.e. reducing three decoders to a shared decoder with task-specific queries and adding an MLP layer to fuse task-specific queries to get coupled features. As Figure 4 shows, the STS paradigm reduces the number of parameters of IterSG[23] and improves its recall rate performance.

## 5.5 Qualitative Results

We visualize the attention map of the last decoder layer for STA baselines (without task-specific queries) and our proposed UniQ as shown in Figure 5. We compare the top 10 relationships predicted

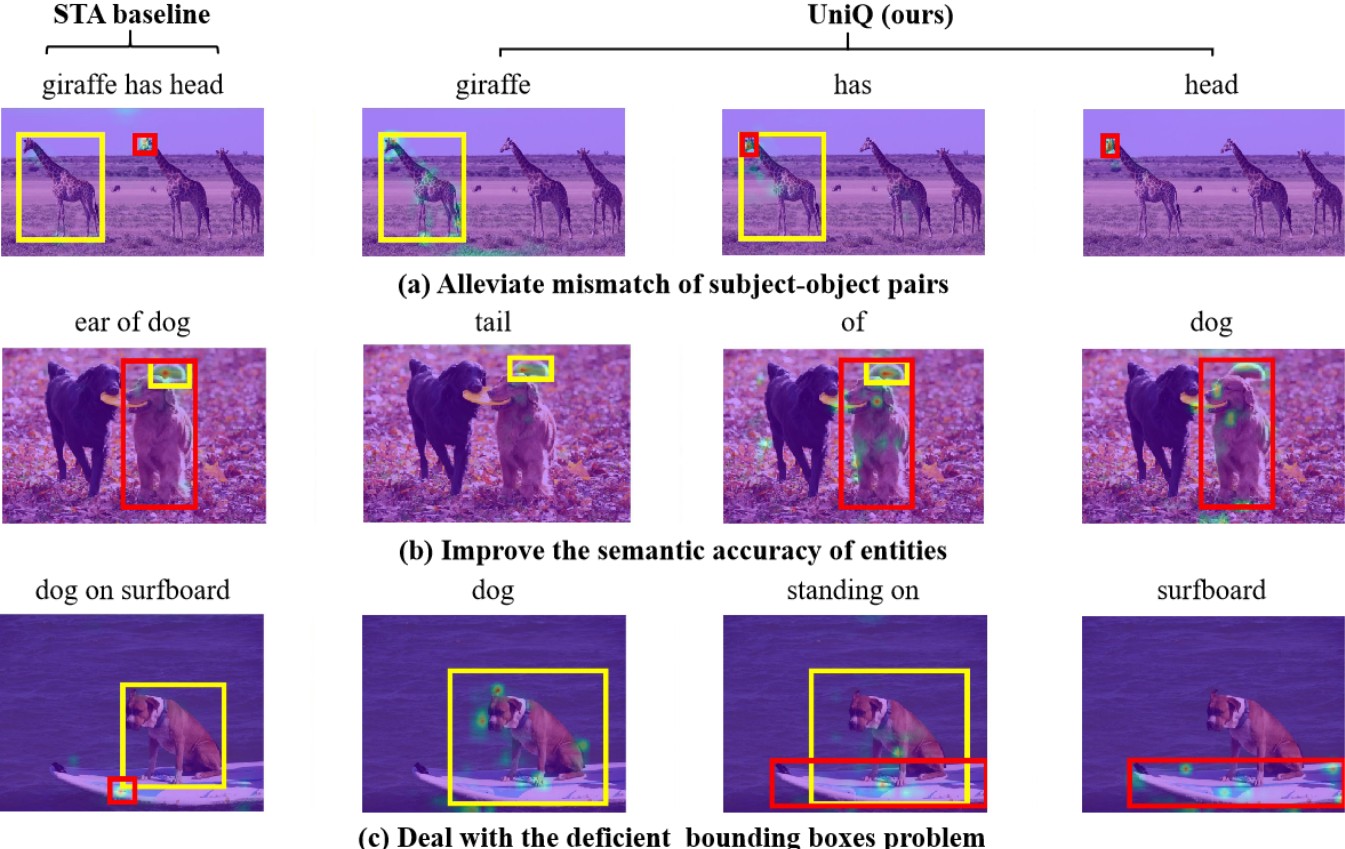

**Figure 5: Qualitative Results. We visualize the decoder's attention map of STA baseline (without task-specific queries) and our UniQ. The first column represents the attention maps for relationships generated by STA baseline. The second to fourth columns represent the attention maps for subjects, predicates, and objects respectively. The yellow rectangles denote the bounding boxes of subjects and the red rectangles denote the bounding boxes of objects.**

by them and find out our proposed UniQ has the ability to deal with the following problems that exist in STA baselines.

**Alleviate mismatch of subject-object pairs.** According to the first row in Figure 5, the STA baseline matched the giraffe in the yellow rectangle with another giraffe's head, while UniQ matched the giraffe with its head correctly. This may be due to UniQ facilitating the interaction within triplets that results in more consistent triplets.

**Improve the semantic accuracy of entities.** In the second row of Figure 5, the STA baseline misidentified the dog's tail for its ear, while UniQ predicted the subject as 'tail' accurately. The results prove that task-specific queries adopted by UniQ carry more detailed and accurate semantic representations compared to methods implemented with task-agnostic queries.

**Deal with the deficient bounding boxes problem.** In the last row of Figure 5, the STA baseline predicted the object 'surfboard' with a tiny bounding box, which is not correct, while UniQ can locate the object 'surfboard' more precisely. Additionally, UniQ predicted the predicate as 'standing on' rather than 'on' (predicted by STA baseline), which contains more semantic information. The

results also demonstrate that task-specific queries facilitate more robust representations.

## 6 Conclusions

In this work, we propose a novel one-stage architecture constructed by a unified decoder with task-specific queries (UniQ) for efficient SGG. Our proposed method provides an available solution to the weak entanglement problem in relational triplet prediction. We leverage task-specific queries to locate entities separately and fuse semantic features within triplets to share coupled features. Extensive experiment results show that UniQ surpasses existing one-stage methods and two-stage methods with fewer parameters and prove the effectiveness of UniQ.

## Acknowledgments

This work was supported in part by the National Natural Science Foundation of China under Grant No. 62276110, No. 62172039 and in part by the fund of Joint Laboratory of HUST and Pingan Property Casualty Research (HPL).

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
