# OpenReview forum: "UniQ: Unified Decoder with Task-specific Queries for Efficient Scene Graph Generation"
_acmmm.org/ACMMM/2024/Conference — MM2024 Poster_

### Official Review · Reviewer_iKtk · 2024-05-08

**Rating:** 3
**Confidence:** 1

**Summary:**

This paper introduces a novel approach to Scene Graph Generation (SGG), called UniQ, which aims to identify object entities and reason their relationships within images. Unlike prevailing two-stage methods, UniQ is a one-stage method that integrates learnable queries to jointly reason relational triplets. The paper addresses the challenge of weak entanglement in one-stage methods by introducing task-specific queries to model decoupled spatial locations and facilitate interaction within each triplet to model coupled semantic features. UniQ employs a unified decoder with task-specific queries for parallel triplet decoding, enabling detailed representation modeling for each sub-task while reducing parameter overhead compared to methods using multiple decoders. Experimental results on the Visual Genome dataset demonstrate that UniQ outperforms both one-stage and two-stage methods in terms of performance while utilizing fewer parameters.

**Strengths:**

Unified Decoder with Task-Specific Queries: UniQ introduces a unified decoder architecture with task-specific queries, allowing for detailed representation modeling for each sub-task. This approach enhances the flexibility and efficiency of scene graph generation.
Reduction in Parameter Overhead: Compared to methods utilizing multiple decoders, UniQ reduces parameter overhead. This reduction in parameters contributes to improved computational efficiency and model scalability.
Effective Handling of Weak Entanglement: UniQ effectively addresses the challenge of weak entanglement between entities and predicates by employing task-specific queries to model decoupled spatial locations and facilitate interaction within each triplet. This enables more accurate modeling of coupled semantic features.

**Limitations:**

The paper lacks attention to detail, for example, there are issues with the bold annotations in Table 1.
The improvement in results is not significant.
There is insufficient innovation in the method.

**Suitability:**

3

---

### Official Review · Reviewer_NonT · 2024-06-03

**Rating:** 3
**Confidence:** 4

**Summary:**

This paper proposes a new one-stage method, UniQ, for scene graph generation tasks. By adopting a unified decoder to extract features from subject-verb-object, UniQ efficiently addresses the weak correlation. It achieves this by using task-specific queries to model spatial locations independently and facilitating interaction within each triplet to model correlated semantic features. It not only significantly reduces computational costs but also improves prediction results

**Strengths:**

1. Introduces a new paradigm for scene graph generation called Single Decoder with Task-Specific Queries (STS). By using a unified decoder for the subject-verb-object triplets, it not only significantly reduces computational costs but also improves prediction results.
2. Partially solves the challenge in one-stage methods arising from weak entanglement.
3. The STS paradigm appears to be simple and easy to transfer, providing reference for other methods in scene graph generation tasks.

**Limitations:**

1. **Insufficient experiments**: The experimental section only includes simple comparisons and a small number of ablation experiments. The paper claims that the STS paradigm can simultaneously capture both coupled features shared within triplets and decoupled visual features, but there is no supporting evidence in the experimental section. Rational experiments can further substantiate the viewpoints of a thesis and make it more comprehensive.
2. **Lack of explanation for the STA and TTS paradigm methods:** Although there is a brief explanation of the STA and TTS paradigms in the third chapter, there are no specific references to the corresponding methods. Additionally, Table 1 only provides classification of encoders, lacking categorization and comparison of relevant paradigms.
3. **The transferability of the STS paradigm has not been discussed**. It is suggested to supplement experiments that demonstrate the transferability of the STS paradigm, which would also serve as evidence for its ability to capture both coupled features shared within triplets and decoupled visual features to some extent.

**Suitability:**

3

---

### Official Review · Reviewer_5gCL · 2024-06-05

**Rating:** 4
**Confidence:** 3

**Summary:**

This work introduces a one-stage SGG formulation. The authors design a unified decoder with task-specific queries architecture, where task-specific queries generate decoupled visual features for subjects, objects, and predicates respectively. Experiments conducted on VG dataset demonstrate the effectiveness of the proposed method.

**Strengths:**

- The authors analyze the challenge of one-stage SGG methods and introduce a unified decoder with a task-specific queries architecture into the SGG task.
- The proposed method shows advantages on zero-shot and no-graph constraint SGG evaluation, indicating its ability to handle unseen relationships.
- The proposed method uses less parameters and achieves better results compared to one-stage SGG methods.

**Limitations:**

- Although this work has achieved promising results, it is difficult to say that “the proposed method has superior performance to both one-stage and two-stage methods”, since the authors have not comprehensively compared it with all two-stage methods, such as [10], which is cited but not compared in this work. The latest two-stage SGG method compared in this work was published in 2021. I strongly suggest that the authors use a more nuanced way to describe the performance.
- The representation of Algorithm 1 is unclear, making it difficult to understand its function. More detailed explanations of Algorithm 1 are needed.
- The overall presentation of the work requires significant enhancement. Especially in the method part, the motivations for some components are not clearly explained.

**Suitability:**

3

---

### Meta-Review · Area_Chair_SSfP · 2024-07-01

**Recommendation:** Accept (Poster)
**Confidence:** 4

**Metareview:**

This paper presents a framework of unified decoder with task-specific queries for efficient scene graph generation. It receives one borderline accept and two borderline reject. The response well addresses the reviewers' concerns. All reviewers give borderline accept ratings after the rebuttal. The merits, including the new paradigm, less parameters, and good results, are well recognized by the reviewers. Please also incorporate the analysis and results in response into the camera-ready version. Overall, I think the current manuscript meets the requirements of this top conference.